# Physiological Comparison of Two Salt-Excluder Hybrid Grapevine Rootstocks under Salinity Reveals Different Adaptation Qualities

**DOI:** 10.3390/plants12183247

**Published:** 2023-09-13

**Authors:** Pranavkumar Gajjar, Ahmed Ismail, Tabibul Islam, Ahmed G. Darwish, Md Moniruzzaman, Eman Abuslima, Ahmed S. Dawood, Abdelkareem M. El-Saady, Violeta Tsolova, Ashraf El-Kereamy, Peter Nick, Sherif M. Sherif, Michael D. Abazinge, Islam El-Sharkawy

**Affiliations:** 1Center for Viticulture and Small Fruit Research, College of Agriculture and Food Sciences, Florida A&M University, Tallahassee, FL 32308, USA; pranavkumar1.gajjar@famu.edu (P.G.); ahmed.ismail@agr.dmu.edu.eg (A.I.); ahmed.darwish@famu.edu (A.G.D.); md.moniruzzaman@famu.edu (M.M.); violeta.tsolova@famu.edu (V.T.); 2Department of Botany and Plant Sciences, University of California Riverside, Riverside, CA 92521, USA; ashrafe@ucr.edu; 3Department of Horticulture, Faculty of Agriculture, Damanhour University, Damanhour 22516, Egypt; 4Alson H. Smith Jr. Agricultural Research and Extension Center, School of Plant and Environmental Sciences, Virginia Tech, Winchester, VA 22602, USA; tabibul@vt.edu (T.I.); ssherif@vt.edu (S.M.S.); 5Department of Biochemistry, Faculty of Agriculture, Minia University, Minia 61519, Egypt; 6Department of Botany and Microbiology, Faculty of Science, Suez Canal University, Ismailia 41522, Egypt; eman_ramadan@science.suez.edu.eg; 7Horticulture Department, Faculty of Agriculture, Al-Azhar University, Cairo 11884, Egypt; ahmed8dawood@azhar.edu.eg; 8Fertilization Technology Department, National Research Center (NRC), Cairo 12622, Egypt; elsaadyam@nrc.sci.eg; 9Molecular Cell Biology, Botanical Institute, Karlsruhe Institute of Technology (KIT), 76131 Karlsruhe, Germany; peter.nick@kit.edu; 10School of the Environment, Florida A&M University, Tallahassee, FL 32307, USA; michael.abazinge@famu.edu

**Keywords:** salinity, grapevines rootstocks, photosynthesis, ROS detoxification machinery, sugar accumulation

## Abstract

Like other plant stresses, salinity is a central agricultural problem, mainly in arid or semi-arid regions. Therefore, salt-adapted plants have evolved several adaptation strategies to counteract salt-related events, such as photosynthesis inhibition, metabolic toxicity, and reactive oxygen species (ROS) formation. European grapes are usually grafted onto salt-tolerant rootstocks as a cultivation practice to alleviate salinity-dependent damage. In the current study, two grape rootstocks, 140 Ruggeri (RUG) and Millardet et de Grasset 420A (MGT), were utilized to evaluate the diversity of their salinity adaptation strategies. The results showed that RUG is able to maintain higher levels of the photosynthetic pigments (Chl-T, Chl-a, and Chl-b) under salt stress, and hence accumulates higher levels of total soluble sugars (TSS), monosaccharides, and disaccharides compared with the MGT rootstock. Moreover, it was revealed that the RUG rootstock maintains and/or increases the enzymatic activities of catalase, GPX, and SOD under salinity, giving it a more efficient ROS detoxification machinery under stress.

## 1. Introduction

Plant stress is a central problem in agriculture, resulting in estimated global crop production losses of 65–87% [1]. Soil salinity, which affects more than 833 million hectares worldwide, mainly in arid or semi-arid regions, is of particular interest [2]. The adverse consequences of salinity include a substantial decline in water quality and soil biodiversity that raises the soil degradation rate. Under such circumstances, roughly 20–50% of irrigated land is classified as highly saline, resulting in estimated annual economic costs of more than $27 billion [2,3]. Salinity mainly arises due to high levels of sodium (Na^+^) and chloride (Cl^−^) ions in the soil’s water-soluble fraction, leading to hyperionic and hyperosmotic stresses. It impairs key plant biological processes, including water and nutrient acquisition, photosynthesis, amino acid and protein synthesis, and energy and lipid metabolism. Subsequently, several salt-related events occur within stressed plants, such as membrane disorganization, metabolic toxicity, reactive oxygen species (ROS) formation, and photosynthesis inhibition [4,5].

Plants have evolved different adaptation strategies to counteract the detrimental impacts of salt stress and are therefore categorized—based on their adaptability—as halophytes and glycophytes [6]. Halophytes are salt-adapted and exhibit a high capacity to cope with stable salty environments. In contrast, glycophytes are salt-sensitive and are limited to inhabiting low-sodium ecosystems [7]. Salt adaptation mechanisms occur at the cellular, molecular, and organismic (whole-plant) levels. For instance, NaCl-stressed plants safeguard ion homeostasis via ion exclusion, compartmentalization, and selective accumulation mechanisms [8,9,10]. In the same manner, plants have to maintain their osmotic balance and redox homeostasis. Osmotic adjustment is achieved by synthesizing and accumulating a wide range of compatible solutes, such as betaines, proline, amino acids, and sugar alcohols [11]. ROS (e.g., H_2_O_2_ and O_2_^−^) are inevitable by-products of aerobic metabolism (e.g., photosynthesis, photorespiration, and respiration) that are generated in several cellular organelles (e.g., chloroplasts, mitochondria, and peroxisomes) [12]. However, ROS are a common feature of abiotic and biotic stress-associated events [13]. They perform a dual action based on their levels and production sites [14]. ROS act as essential signaling molecules at low levels, while excessive ROS ultimately leads to cell death [14,15]. Therefore, plants utilize different enzymatic and non-enzymatic antioxidative defense mechanisms that strictly restrain ROS levels. Catalase (CAT), glutathione peroxidase (GPX), ascorbate peroxidase (APX), and superoxide dismutase (SOD) are examples of the enzymatic antioxidant mechanisms. However, proline (Pro), glutathione (GSH), ascorbic acid (ASH), phenolic compounds, and alkaloids are examples of non-enzymatic ROS-scavenging machinery [14,16].

Bunch grapes (*Euvitis*), mainly *Vitis vinifera*, are considered among the most economically important fruit species worldwide. Although grapes are moderately salt-tolerant, they are more sensitive to Cl^−^ toxicity than Na^+^ [17,18]. Grapes are ideally cultivated in hot and semi-arid regions and hence are challenged by different types of water-related stresses, such as salinity, which affects viticulture, impairing grape production and quality [19,20]. To alleviate salinity-dependent damage, European grapes are grafted onto salt-tolerant rootstocks, such as those derived from winter grape (*V. berlandieri*) and sand grape (*V. rupestris*) [17,21]. The current research aimed to evaluate the different effects of salinity on two grape rootstocks, 140 Ruggeri (RUG) and Millardet et de Grasset 420A (MGT). The two rootstocks are salt-excluders, exhibiting effective NaCl-exclusion capacity that reduces salt accumulation in the leaves and berries of grafted *V. vinifera* scions [20,22]. However, salt-exclusion capacity is not the only determining factor of salinity adaptation [15,23,24]. In this study, we performed a comparative physiological analysis of these two rootstocks (RUG and MGT) under salinity and during recovery. Our research objective was to examine the diversity of the physiological responses of RUG and MGT under gradually increased salinity and during recovery and thereby come to understand the integral mechanisms that function in parallel with salt exclusion in both rootstocks to alleviate salt-induced damage. We measured photosynthetic pigments (Chl-T, Chl-a, and Chl-b) and sugars as an indication of efficient photosystem activity and sugar/energy metabolism, respectively. Moreover, redox homeostasis was investigated by evaluating the enzymatic activities of CAT, GPX, and APX, as well as the content of Pro, along with two oxidative stress markers, malondialdehyde (MDA) and H_2_O_2_.

## 2. Results

### 2.1. Changes in Pigment Content in Response to Salt Stress

Two salt-excluder grapevine rootstocks, RUG and MGT, which differ in their adaptability to salinity, were utilized to gain comprehensive insights into their physiological responses to salt stress. The two rootstocks were watered daily with 30 mM NaCl solution for 5 days. This procedure of gradual salt increase was used to avoid potential responses due to osmotic shock [25]. Once the treated rootstocks had received their fifth, final dose of 30 mM NaCl on day 5 of the treatment, we assumed that the salt concentration in the soil solution was roughly 150 mM NaCl, and hence leaf samples were collected at 0.5, 2, 24, and 48 h. A sample was gathered at time zero (time-0) before the fifth supplementation of 30 mM NaCl, while the control plants did not receive any salt treatment. For the recovery experiment, both rootstocks were re-watered with potable water, and leaf samples were collected at 4 and 12 days.

An evaluation of photosynthetic pigments revealed that the RUG and MGT rootstocks exhibited discernible patterns, although both are salt-excluders [21] (Figure 1). Under typical growth conditions (control), the basal levels of total chlorophyll (Chl-T) and chlorophyll-a (Chl-a) were comparable in both rootstocks, while chlorophyll-b (Chl-b) was significantly higher in the RUG leaves (Figure 1A–C). The RUG rootstock showed higher levels of photosynthetic pigments under gradually increased salinity compared with the MGT rootstock (~1.35–~1.95-fold at time-0). Generally, Chl-T, Chl-a, and Chl-b displayed similar accumulation patterns in the salt-stressed RUG (Figure 1A–C). The significant increase in photosynthetic pigments at time-0 remained stable during the short-term salt stress (0.5–2 h) before increasing again during the long-term stress (24 h–48 h). By contrast, the photosynthetic pigments of the salt-stressed MGT showed weak induction patterns. The salt-stressed MGT also exhibited a delayed response; significant accumulations of different chlorophyll components were observed after 2 h of the last NaCl application compared to the control. Interestingly, all the photosynthetic pigments dropped during the recovery time in both rootstocks, returning to their original control levels.

In contrast, the amount of total carotenoids observed in the MGT was roughly twice that observed in the RUG under typical growth conditions. However, under gradually increasing salinity, the carotenoid levels significantly dropped in both rootstocks (at time-0; Figure 1D). Interestingly, the RUG was able to restore its initial carotenoid levels after 2 h of the salt addition process and maintain them for 24 h before a second drop at 48 h. Perplexingly, this was not the case for the MGT, which showed insignificant changes in its carotenoid levels during 48 h of salt stress. During recovery, both rootstocks displayed similar amounts of carotenoids; these levels were equal to the basal levels in the RUG, but not in the MGT (Figure 1D). In general, the Chl-T, Chl-a, and Chl-b showed evident salt-dependent trends in the RUG, but not in the MGT (Figure 1A–C).

### 2.2. Under Salt Stress, RUG Accumulated More Soluble Sugars Than MGT

Sugars are considered among the most crucial regulators of many physiological processes, including photosynthesis, flowering, and senescence, as well as responses to different abiotic stresses [26]. In both rootstocks, the total soluble sugar (TSS), glucose (Glu), fructose (Fru), and sucrose (Suc) contents showed roughly inconsistent patterns throughout the salinity treatment and during recovery (Figure 2). For instance, the TSS significantly decreased in the RUG under gradually increasing salinity, then stabilized at around 5 mg/g FW (from time-0 to 2 h post-treatment), before increasing to its initial levels at 48 h, and subsequently dropping again during the recovery course. In contrast, the TSS in the MGT showed no significant change under the salinity treatment and during recovery, except when a slight reduction occurred at 12 d of recovery (Figure 2A).

Similarly, the levels of Glu and Fru decreased throughout progressive salinity increase in the RUG (until time-0, Figure 2B,C). Interestingly, both monosaccharides re-established their initial levels at 48 h of salinity despite their contradictory patterns during the first 24 h post salt deposition. During the recovery time, the levels of both Glu and Fru declined and showed different kinetics. Overall, the TSS and Fru showed roughly similar patterns in the RUG under salinity and during recovery (Figure 2A,C). On the other hand, the patterns of Glu and Fru were comparable in the MGT, exhibiting fluctuations during the stress and recovery times (Figure 2B,C).

Finally, the level of the sucrose (Suc), which is composed of glucose and fructose subunits, showed different kinetics in both rootstocks (Figure 2D). In the RUG, Suc accumulation did not considerably alter during the salt treatment, but then its level sharply declined and increased at 4 d and 12 d of recovery, respectively. In the MGT, however, Suc levels showed a severe reduction throughout the gradual increase in salinity, and this was followed by a slight increase within the first 24 h after salt supplementation and a subsequent decrease at 48 h of stress (Figure 2D). During the subsequent recovery days, the Suc content slightly re-elevated, but it did not reach its initial level. Our data revealed that the RUG accumulated higher levels of TSS, monosaccharides, and disaccharides at late stages of salinity than the MGT.

### 2.3. Under Salinity, RUG Maintained More Efficient Redox Homeostasis Than MGT

To establish an association between the diversity in salinity adaptation and the ability to scavenge ROS, markers for both oxidative damage and redox homeostasis were monitored during the salinity and recovery courses in both rootstocks (Figure 3). For oxidative damage, the content of the well-established oxidative stress biomarker malondialdehyde (MDA) was measured (Figure 3A). Both the RUG and MGT showed comparable MDA levels under salinity treatment and during recovery, with few exceptions. In the RUG, the level of MDA during salinity and recovery remained constant at around the control level. However, in the MGT, the initial level of MDA was twice as high as that found at time-0 and during the first 24 h of salt addition (0 h–24 h; Figure 3A). It then increased significantly at 48 h of salinity and during recovery, and during this time it was consistent with the MDA level in the RUG (48 h and 4 d–12 d; Figure 3A). These data indicate that MDA cannot be considered as a stress marker in the two rootstocks under our experimental conditions. Both rootstocks showed a substantial increase in hydrogen peroxide (H_2_O_2_) content, the most stable ROS, during the first 0.5 h of salt addition (~1.8-fold). However, the H_2_O_2_ accumulation patterns of the two rootstocks started to differentiate at 2 h of salinity, increasing significantly in the MGT, but reducing in the RUG (Figure 3B). In general, the H_2_O_2_ patterns were inconsistent between the two rootstocks, showing higher levels at 2 h and 24 h of salinity, as well as at 4 d of recovery in the MGT than in the RUG at the same time points. In summary, our data show that the RUG was more efficient in constraining the increase in H_2_O_2_ under salt stress than the MGT.

For redox homeostasis, plants utilize a battery of enzymatic and non-enzymatic antioxidants. Interestingly, the levels of the non-enzymatic antioxidant proline overlapped in both the RUG and the MGT, which exhibited a strong increase in proline content only at 48 h of salinity (Figure 3C). The data indicate that proline is not a distinctive adaptation factor in either of the two rootstocks. On the other hand, the patterns of the enzymatic antioxidants (e.g., catalase, GPX, and SOD) were perplexing (Figure 3D–F). For instance, the catalase activity showed insignificant changes in the RUG leaves at time-0 and during the first 48 h of salt stress (0 h–48 h; Figure 3D) before starting to drop significantly, reaching its basal level at day 4 of recovery, but being restored at 12 d (Figure 3D). In contrast, catalase activity in the MGT leaves exhibited varied pattern under salinity. For instance, catalase activity declined considerably in the salt-stressed MGT leaves at time-0 (~43%), then gradually increased by up to ~1.8-fold relative to the control at 2 h post salt deposition, before stabilizing roughly around its basal levels at 48 h and during the recovery process (Figure 3D). The GPX activity, however, slightly declined in the salt-stressed RUG before being restored to its initial control levels at 48 h of salinity and 12 d of recovery (Figure 3E). In contrast, the MGT leaves exhibited a robust and sharp reduction in their GPX activity compared with those of the control plants during the first 24 h of NaCl application (~83% reduction). The GPX activity in the MTG then partially recovered at 48 h, reaching levels similar to those of its counterpart, the RUG (Figure 3E). Interestingly, the GPX activity patterns in the RUG and MGT during the recovery process were inverted. The SOD activity pattern was roughly similar to that of GPX activity in both rootstocks, but with different kinetics (Figure 3F). In the RUG, SOD activity did not significantly respond to salt stress before showing a ~1.5-fold increase at 48 h of salinity compared with the levels observed in the control and those observed at previous salt-stressed time points. In the salt-stressed MGT, however, the SOD activity pattern was nearly comparable to the GPX activity pattern (Figure 3E,F). This was also the case during the recovery process, during which the GPX and SOD activity patterns were similar.

## 3. Discussion

The present study aimed to discriminate the salt adaptation strengths of two salt-excluder grapevine rootstocks, RUG and MGT. However, because of its drought adaptation capacity, RUG was expected to have a more efficient salt adaptation mechanism. It is well documented that excessive salt accumulation in plant cells results in many physiological disorders, including photosynthesis inhibition. Therefore, salt-tolerant plants have to maintain high photosynthetic efficiency under salinity. Indeed, the RUG exhibited a significant accumulation of Chl-T, Chl-a, and Chl-b, and its induction patterns were roughly the same as those of the MGT under gradually increasing salinity. Similarly, salt-excluder grapevine rootstock 1103 Paulsen maintained higher chlorophyll levels than the MGT rootstock [27]. These elevations in Chl-T, Chl-a, and Chl-b in the RUG vanished during the recovery process in both rootstocks, indicating an apparent association between the robust increases in the photosynthetic pigments and the salinity adaptation qualities of the RUG (an association that was weaker in the MGT). Obviously, the photosynthetic pigments convert solar energy into chemical energy (sugars) and thereby govern the growth and development of higher plants [5,28]. Interestingly, the TSS levels differed significantly in both rootstocks under non-stressful conditions; however, the RUG accumulated higher amounts of monosaccharides (Glu and Fru) than the MGT, which maintained more elevated levels of disaccharides (Sac). Unexpectedly, the induction of photosynthetic pigments did not result in a pronounced accumulation of sugars to a level reflective of the significant increases in Chl-T, Chl-a, and Chl-b in the RUG during the first hours of salinity. On the contrary, the levels of TSS, Glu, and Fru decreased throughout the stress progression in the RUG before returning to their initial levels during long-term stress. This could be attributed to the reduction in stomatal conductance during salinity, which deprived the chloroplasts of atmospheric CO_2_ and hence lowered the rate of photosynthetic carbon assimilation [29,30]. In addition, the metabolomic profiling of the Cabernet grapevine under salinity showed that the abundance of 22 sugar metabolites was significantly differentiated, of which Glu was decreased, while Fru was unchanged [29]. In fact, low sugar levels promote photosynthesis and reserve mobilization and export, whereas high sugar accumulation leads to carbohydrate storage and growth [31]. However, under stress, plants have to prioritize defense over growth and development [32].

Like other plant stresses, salinity results in increased free radicals and, subsequently, MDA accumulation. The overproduction of MDA and electrolyte leakage are symptoms of membrane damage stemming from salinity [33]. Interestingly, this was not the case in the current study, as both rootstocks showed comparable levels of MDA under salinity and during the recovery process. However, the H_2_O_2_ content increased significantly during the gradual application of salt in both rootstocks, and a distinctive pattern emerged after 0.5 h of stress. While MGT stabilized the levels of H_2_O_2_, the RUG was able to effectively decrease its H_2_O_2_ levels, particularly at 2 h of stress and at 4 d of recovery. Apparently, both rootstocks have evolved efficient ROS scavenging machinery, but different adaptation qualities. Therefore, the capacity of non-enzymatic antioxidants (proline content) and the enzymatic activity profiles of CAT, GPX, and SOD were examined in both rootstocks. In addition to its antioxidant contribution, proline plays several physiological and biochemical roles, such as osmoprotection and photosynthesis improvement [34,35]. However, proline has been excluded from being a distinctive adaptation factor, despite its substantial accumulation in the RUG and MGT leaves after 48 h of salt deposition.

On the other hand, the enzymatic activities of CAT, GPX, and SOD may represent plausible distinguishing factors, although they showed perplexing patterns in both rootstocks under salinity. The lower activities of GPX and SOD (and to some extent CAT) in the MGT compared with the RUG during the first 24 h of the gradual increase in salinity suggest that both rootstocks differ mainly in their capacity to maintaining their antioxidant systems under stress. Hence, RUG was more robust in redox homeostasis than MGT under salt stress. However, further analyses are required to elucidate the different adaptive capacity levels of both rootstocks, particularly at the anatomical, molecular, and cellular levels, as well as the involvement of other (non)-enzymatic antioxidants. 

## 4. Materials and Methods

### 4.1. Plant Materials

Leaf samples were collected from 3-year-old *Vitis* hybrids, 140 Ruggeri (140 Ru; *V. berlandieri* × *V. rupestris*) and Millardet et de Grasset 420A (MGT; *V. berlandieri* × *V. riparia*), grown at the experimental vineyard of the Florida A&M University (Tallahassee, FL, USA). These two hybrid rootstocks were selected according to their diversity in salinity adaptation [22]. Salt stress was applied gradually by irrigating the plants with 30 mM NaCl daily for five days until a final concentration of 150 mM NaCl was reached. Once the plants received their fifth dose of NaCl, samples were collected from the mature and healthy leaves (3rd to 4th leaf). For the salinity time course, the leaf samples were collected at 0.5, 2, 24, and 48 h after the application of the fifth dose of NaCl. Leaves collected just before the addition of the fifth salt treatment were considered time-zero samples. The control plants were not treated with salt. The salt-treated rootstock plants were flushed with salt-free pure water for the recovery experiment, and leaf samples were collected 4 and 8 days after treatment. All samples were immediately frozen in liquid nitrogen and stored at −80 °C for further analysis.

### 4.2. Chlorophyll and Carotenoid Contents

The photosynthetic pigment concentrations were quantified as destructive traits. Extraction was carried out using dimethyl sulfoxide (DMSO) solvent [36]. Briefly, 50 mg leaf tissue was homogenized with 1.5 mL DMSO. The reactions were incubated in a water bath at 65 °C for one hour and cooled at room temperature for 30 min. After filtration, the mixture was shaken and the absorbance was measured at λ = 665 for the Chl-a, λ = 648 nm for the Chl-b, and λ = 480 for the β-carotene using a microplate reader (ACCURIS SmartReader; Edison, NJ, USA) blanked with DMSO. Concentrations were estimated for each biological replicate in triplicate (*n* = 9) and expressed as mg/g fresh weight. The Chl-a, Chl-b, and β-carotene contents were calculated using the following equations:Chlorophyll a (mg/g) = 12.47 (A_665_) − 3.62 (A_648_) × V/1000 × W
Chlorophyll b (mg/g) = 25.06 (A_648_) − 6.50 (A_665_) × V/1000 × W
C_x+c_ = (1000 (A_480_) − 1.29Ca − 53.78Cb)/220.

C_x+c_: Concentration of xanthophylls and carotenes; Ca: Chl-a; Cb: Chl-b.

### 4.3. Quantification of Reactive Oxygen Species (ROS) Content

The reactive oxygen species were assessed according to the method outlined by Islam et al. [37]. Briefly, 1 mL of 50 mM KPO_4_^−^ buffer (pH 7.8) was added to 100 mg of fresh powdered leaf tissue. The mixture was centrifuged at 12,000× *g* for 10 min at 4 °C. To determine the H_2_O_2_ levels, the extracted solution was mixed with 0.1% titanium chloride in 20% (*v*/*v*) H_2_SO_4_ and then centrifuged at 10,000× *g* for 5 min. The absorbance was measured at λ = 410 nm using a microplate reader (ACCURIS SmartReader; Edison, NJ, USA). The H_2_O_2_ level was calculated using an extinction coefficient of 0.28 μmol^−1^ cm^−1^.

### 4.4. Lipid Peroxidation Content

Lipid peroxidation content was assessed in relation to MDA production in the tissue samples using the commercially available Lipid Peroxidation (MDA) Assay Kit (Abcam, Waltham, MA, USA). The assay was performed according to the protocol, along with the optional step for enhanced sensitivity. Briefly, 0.1 g tissue powder was added to 1 mL extracting solution. The mixture was centrifuged at 8000× *g* for 10 min at 4 °C. Next, 0.2 mL of the supernatant was removed, mixed with 0.6 mL of TBA, and moderately shaken. The reactions were incubated at 95 °C for 30 min and then centrifuged at 10,000× *g* for 10 min at 25 °C. The absorbance was measured at λ = 532 and 600 nm using a microplate reader (ACCURIS SmartReader; Edison, NJ, USA) to determine the MDA concentration [38].

### 4.5. Determination of Proline Content

The proline content was measured according to the method outlined by Lee et al. [39], though with a slight modification. Briefly, 100 mg of fresh powdered leaf tissue was extracted using 3% sulfosalicylic acid. After centrifugation, the supernatants were mixed with a ninhydrin solution containing acetic acid and 6 M H_3_PO_4_ (*v*/*v*, 3:2) and boiled at 100 °C for 60 min. Toluene was then added to the mixture, which was subsequently incubated for 30 min at room temperature. The absorbance was determined at λ = 520 nm using a microplate reader (ACCURIS SmartReader; Edison, NJ, USA) and calculated using L-proline (1–100 μg).

### 4.6. Soluble Sugar Analyses

The soluble sugars were extracted according to the method outlined by Islam et al. [40]. Briefly, the soluble sugars were extracted from 100 mg samples of the fresh ground leaf tissue using 1 mL of 80% ethanol. This was followed by vortexing and centrifugation at 12,000× *g* for 10 min. The supernatant was collected, and this extraction step was repeated twice. The glucose, sucrose, and fructose contents were assessed using the Megazyme Sucrose/D-Fructose/D-Glucose Assay Kit (Megazyme, Highland, UT, USA) in accordance with the manufacturer’s protocol. The total soluble sugar was determined by adding up the glucose, fructose, and sucrose content values of in each sample.

### 4.7. Antioxidative Enzymes Activity Assays

To quantify the antioxidant enzymes, 100 mg of fresh ground leaf tissue was homogenized in 50 mM potassium phosphate buffer (pH 7.0) and centrifuged at 12,000× *g* for 10 min. The supernatant was used for the measurement of the total protein content and for the enzyme activity assays. The total protein was quantified according to the Bradford assay using bovine serum albumin (BSA) as a standard. The activity of the superoxide dismutase (SOD), catalase (CAT), and glutathione peroxidase (GPx) enzymes was assayed according to the method outlined by Cavalcanti et al. [41] using assay kits from BioVision Inc. (Milpitas, CA, USA) in accordance with the manufacturer’s instructions. The SOD activity was expressed as inhibition rate (%) per mg of protein. One unit of CAT was expressed as the amount of CAT that decomposed 1.0 nmol of H_2_O_2_ min^−1^. The amount of GPx that caused a decrease of 1.0 nmol of NADPH min^−1^ was expressed as one unit.

### 4.8. Statistical Analyses

The statistical analyses were performed via a multivariate ANOVA using IBM SPSS Statistics software (version 22.0). The results were expressed as the mean ± SE of three independent replicates. Different letters indicate significant differences between different treatments and genotypes according to Duncan’s test (*p* < 0.05). The letters were alphabetically ordered in an ascendant manner, where “a” always represented the lowest significant value. Values with same letters were not significantly different. 

## 5. Conclusions

This study demonstrated, on a physiological level, the differences in the salinity adaptation mechanisms of two grapevine rootstocks. Although RUG and MGT are considered salt-excluder rootstocks, the highly tolerant rootstock RUG showed a greater capacity for maintaining the photosynthetic pigments (Chl-T, Chl-a, and Chl-b) under salinity, and subsequently accumulated higher amounts of TSS, monosaccharides, and disaccharides as a source of energy, compared with the MGT rootstock. In addition, the ROS detoxification machinery seemed to be more efficient in the RUG than in the MGT under salt stress. Therefore, the adoption of such salt-excluder rootstocks as RUG in commercial vineyards situated in arid or semi-arid regions would help improve grapevine productivity, paving the way for sustainable agriculture under salinity stress. 

## Figures and Tables

**Figure 1 plants-12-03247-f001:**
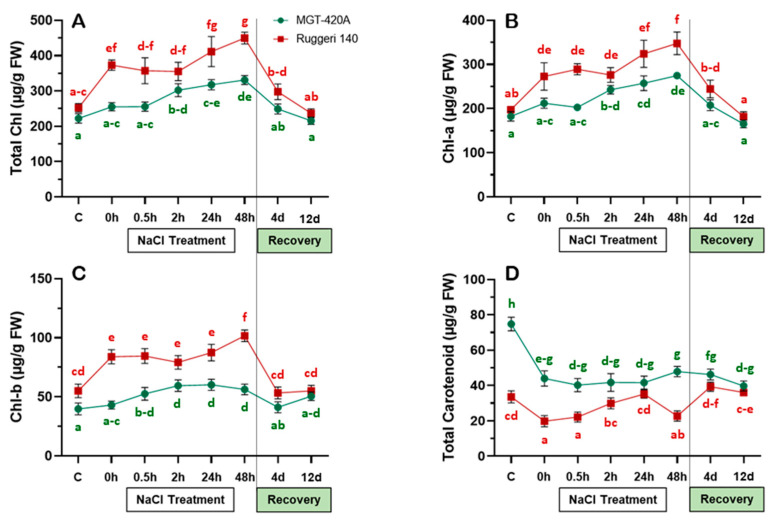
Chlorophyll and carotenoid contents of the two grapevine rootstocks during salt treatment and recovery. (**A**) Total chlorophyll (Chl-T); (**B**) chlorophyll a (Chl-a); (**C**) chlorophyll b (Chl-b); and (**D**) total carotenoid contents of Ruggeri 140 (RUG) and MGT-420A (MGT) grapevine rootstocks subjected to salt treatment and recovery. Values represent the mean of three biological replicates ± standard error (SE). The letters were alphabetically ordered in an ascendant manner, where “a” always represented the lowest significant value. Values with same letters were not significantly different.

**Figure 2 plants-12-03247-f002:**
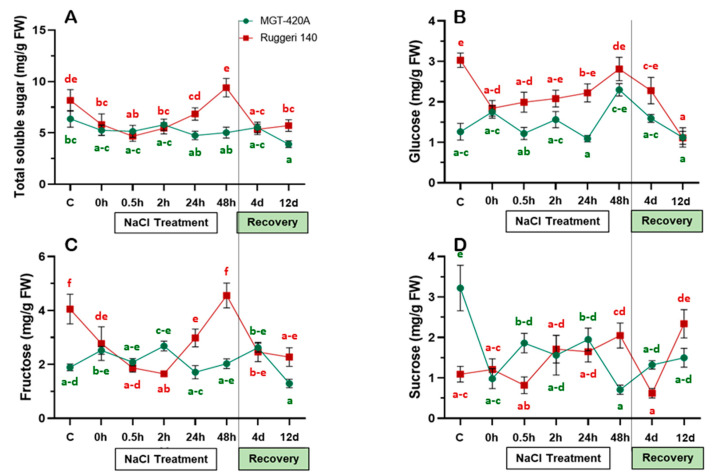
Soluble sugar contents of the two grapevine rootstocks during salt treatment and recovery. (**A**) Total soluble sugar (TSS); (**B**) glucose (Glu); (**C**) fructose (Fru); and (**D**) sucrose (Suc) contents of Ruggeri 140 (RUG) and MGT-420A (MGT) grapevine rootstocks subjected to salt treatment and recovery. Values represent the mean of three biological replicates ± standard error (SE). The letters were alphabetically ordered in an ascendant manner, where “a” always represented the lowest significant value. Values with same letters were not significantly different.

**Figure 3 plants-12-03247-f003:**
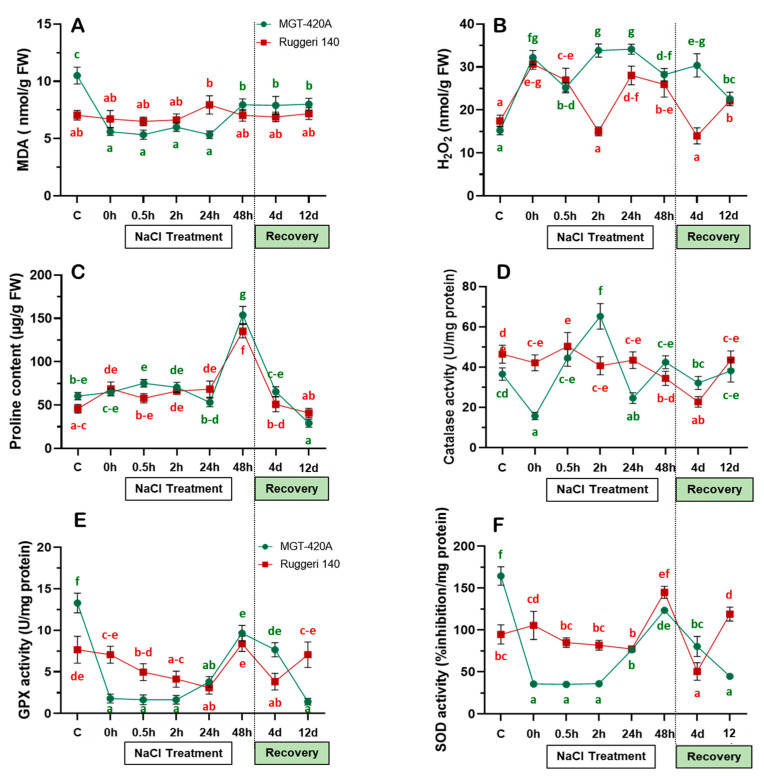
Contents of different ROS-related compounds in the two grapevine rootstocks during salt treatment and recovery. (**A**) Malondialdehyde (MDA); (**B**) H_2_O_2_; (**C**) proline; (**D**) catalase; (**E**) glutathione peroxidase (GPX); and (**F**) superoxide dismutase (SOD) contents of Ruggeri 140 (RUG) and MGT-420A (MGT) grapevine rootstocks during salt treatment and recovery. Values represent the mean of three biological replicates ± standard error (SE). The letters were alphabetically ordered in an ascendant manner, where “a” always represented the lowest significant value. Values with same letters were not significantly different.

## Data Availability

The data presented in this study are available to anyone on request.

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
