# Peer review of "Physiological Comparison of Two Salt-Excluder Hybrid Grapevine Rootstocks under Salinity Reveals Different Adaptation Qualities"

_plants, 2023, doi:10.3390/plants12183247_

Round 1

Reviewer 1 Report

Ms. Ref. No.: plants-2541138
Title: Physiological Comparison of Two Salt-Excluder Hybrid Grapevine Rootstocks Under Salinity Reveals Differential Adaptation Quality

Plants
Dear Dr. Gajjar
,
I am very sorry to inform you that your manuscript is not suitable for the publication in Plants as it stands. The research results reported are weak and descriptive and do not provide sufficient information about the mechanisms adopted by the grapevines rootstocks to cope with stress conditions. 

Author Response

Response to reviewer's comments:

# Reviewer 1:

  • The research results reported are weak and descriptive and do not provide sufficient information about the mechanisms adopted by the grapevine rootstocks to cope with stress conditions.

-We appreciate the reviewer’s comment. The current study discriminates salinity adaptation strength in two salt-excluder grapevine rootstocks, RUG and MGT, following three critical parameters including photosynthesis, sugars, and ROS detoxification. As expected, the highly salt and drought-tolerant rootstock RUG showed better accumulation in photosynthetic pigments under salinity compared to MGT, although the basal levels of total chlorophyll (Chl-T) and chlorophyll-a (Chl-a) were comparable in both rootstocks, but not chlorophyll-b (Chl-b), which was significantly higher in RUG leaves. Unexpectedly, this induction of photosynthetic pigments did not result in a pronounced accumulation of sugars to a proportional level that reflects the significant increase of Chl-T, Chl-a, and Chl-b in RUG during the first hours of salinity, relative to MGT. Indeed, the induction of sugar levels required more time to be significantly visible, with inconsistent patterns among every component (TSS, Glu, Fru, and Suc). Surprisingly, the increase of sugar components under salinity never surpassed its basal levels at any stress point in both rootstocks, especially in RUG, which accumulated higher amounts of photosynthetic pigments under salinity compared to non-saline conditions. In contrast, the initial level of Suc was never achieved in MGT nor during the salinity of re-watering time. Regarding ROS, the content of MDA did not show any significant change in both rootstock and hence could not be considered as a stress marker under our experimental condition. On the contrary, the levels of H2O2 were higher at 2 – 24 h of salinity in MGT, representing a plausible salinity stress marker in our study. Unexpectedly, this induction pattern of proline is highly similar in both rootstocks during stress and recovery. For antioxidant enzyme activities, we expected a significant increase in catalase, GPX, and SOD or at least one of them under salinity in both rootstocks as they are salt-excluder rootstocks with some levels of salinity adaptation. Unfortunately, this was not the case, and their patterns were perplexing. In RUG, for example, their levels of SOD were stabilized during the first 24 h, before increasing at 48 h. The GPX activity showed, however, a slight decline in RUG before restoring its initial control levels at 48h of salinity. In contrast, the activity level of GPX activity sharply dropped at 0h of salinity, before increasing at 48 h, similar to RUG. catalase activity did not exhibit any significant increase in salinity in RUG, although a fluctuation pattern took place in MGT. This data indicated the critical role of maintaining and/or increasing, the enzymatic activity of catalase, GPX, and SOD under salinity, the feature that is more pronounced in RUG relative to MGT. So, we are confident that the physiological story is connected and informative, although we understand that more analysis at different levels is required. Therefore, we are currently analyzing the RNAseq data for salt-stressed RUG, as well as linking these physiological data with the transcriptomic data to further understand the underlying mechanisms.

Reviewer 2 Report

The work presents an interesting, multistep study of the metabolic response attributed to the saline stress in two grape rootstocks. Plant stress is a great problem in agriculture influencing also grape production. As the analysis of the plant stress response mechanisms, as well as the search for salt resistance rootstocks, is significant for the agricultural industry, the topic of the research is valid. The obtained results do not give the complete answer and further research is required, but as giving new insight in comparison to so far published research, this paper would be interesting not only as the basis for further analysis of the Authors but also for other research groups. 

Generally, the study is well designed and interpreted, can arouse public interest, and can consider for publication. I read the manuscript thoroughly and I can not find any serious allegations therefore I would recommend it for publication in Plants journal after addressing minor remarks.

Minor remarks 

Lines 93-99 As the section is an Introduction I would suggest excluding the description of the results and removing it to the appropriate section of the manuscript. One sentence describes the aim of the study, on the other hand.  “What are the other integral mechanisms that empower 93 salinity adaptation in both rootstocks?” It is very general and should be specified for this manuscript. 

Line 277 Leaf not leave

Author Response

Response to reviewer's comments

  • The obtained results do not give the complete answer and further research is required, but as giving new insight in comparison to so far published research, this paper would be interesting not only as the basis for further analysis of the Authors but also for other research groups.

  • We appreciate the reviewer’s comment. In fact, we understand that more analysis at different levels is required, such as at the transcriptomic and genomic levels. Therefore, we are currently analyzing the RNAseq data for salt-stressed RUG, as well as linking these physiological data with the transcriptomic data to further understand the underlying mechanisms.

# Reviewer 2 Minor remarks:

  • Lines 93-99 As the section is an “Introduction”, I would suggest excluding the description of the results and removing it to the appropriate section of the manuscript. One sentence describes the aim of the study, on the other hand. “What are the other integral mechanisms that empower 93 salinity adaptation in both rootstocks?” It is very general and should be specified for this manuscript.

- We appreciate the reviewer’s comment. This part was modified accordingly.

- Line 277 Leaf does not leave.

- The word was changed.

Reviewer 3 Report

The manuscript is devoted to the study of the adaptability of grape on two grape rootstocks to plant stress under salinity conditions. Interesting results worthy of publication in this journal are described. Several recommendations can be made for better understanding and readability of the work.

- It is recommended to move the methods section before the results section, then the procedure of the gradual increase in salt will not be explained twice;

- Discussion and results sections can be merged for better readability;

- The statistical analysis is not described in sufficient detail, especially the meaning of the letters needs description.

Some technical errors:

- The sentence “The moderately salt-tolerant grapes are more sensitive to Cl- toxicity rather than 33 Na+” in the abstract is not further explored and should be removed;

- p.1 r. 41 “lower levels” instead of “lesser levels”;

- 2.3 has the same heading as 2.2. and should be changed;

- The literature [36] does not seem to be relevant;

- p.9 r.328 the term ‘as described previously’ is not precise.

Author Response

Response to reviewer's comments

# Reviewer 3:

Comments and Suggestions for Authors

  • The manuscript is devoted to the study of the adaptability of grapes on two grape rootstocks to plant stress under salinity conditions. Interesting results worthy of publication in this journal are described. Several recommendations can be made for better understanding and readability of the work.

  • We highly appreciate the reviewer’s opinion, and we fully agree with it.

# Reviewer 3 Comments:

  • It is recommended to move the methods section before the results section, then the procedure of the gradual increase in salt will not be explained twice.

  • We appreciate the reviewer’s comment, we followed the journal’s format.
  • Discussion and results sections can be merged for better readability.

  • We appreciate the reviewer’s comment, we also followed the journal’s format. In addition, we feel that our results may need a separate section to be well discussed.

  • The statistical analysis is not described in sufficient detail, especially the meaning of the letters needs description.

  • The meaning of the letters was described accordingly.

# Reviewer 3 Some technical errors:

  • The sentence “The moderately salt-tolerant grapes are more sensitive to Cl- toxicity rather than 33 Na+” in the abstract is not further explored and should be removed.

- The above-mentioned sentence was removed.

- p.1 r. 41 “lower levels” instead of “lesser levels”

- The sentence was corrected.

  • 3 has the same heading as 2.2. and should be changed.
  • The headings have been corrected.
  • The literature [36] does not seem to be relevant.

  • The right citation was added (Heath & Packer, 1968)

  • P.9 r.328 the term ‘as described previously’ is not precise.

  • The sentence was collected, as a procedure was used with a slight modification.

Reviewer 4 Report

1. Provide more details on the experimental setup and methodology used to evaluate the salt adaptation strategies of the two grape rootstocks. Include information on the growth conditions, salinity treatments, and duration of the experiments. This would enhance the reproducibility of the study and allow readers to better understand the experimental design.

2. Introduction needs to be benefited from more literature, and papers have to be cited in it, i.e., I) doi: 10.1128/spectrum.02311-22 II) doi: 10.1039/d2gc02467e

3. Include additional physiological parameters to assess the salt adaptation of the rootstocks. For example, measuring the activity of antioxidant enzymes such as superoxide dismutase (SOD) and catalase (CAT) would provide insights into the ROS detoxification machinery. Additionally, measuring ion content (Na+, Cl-, etc.) and osmolyte accumulation (e.g., proline) would further elucidate the mechanisms underlying salt tolerance.

4. Discuss the implications of the observed differences in photosynthetic pigments and sugar accumulation between the two rootstocks. How do these differences contribute to their respective salt adaptation strategies? Elaborate on the potential influence of these physiological traits on plant growth, development, and overall salt tolerance. Discussion needed to be more comparatively from previous literature i.e., I) https://doi.org/10.3390/polym14142899

5. Consider including gene expression analysis or proteomic profiling to investigate the molecular mechanisms underlying the observed physiological responses. This would provide a more comprehensive understanding of the molecular pathways involved in salt adaptation and enable identification of specific genes or proteins associated with the differential salt tolerance of the rootstocks.

6. Discuss the practical implications of the findings in the context of grape cultivation in salt-affected regions. Highlight the potential benefits of using salt-excluder rootstocks like RUG in improving grapevine productivity and sustainability under salinity stress. Additionally, discuss the limitations and challenges associated with the adoption of these rootstocks in commercial vineyards.

English is okay

Author Response

Response to reviewer's comments:

# Reviewer 4:

We highly appreciate the reviewer’s opinion, and we fully agree with it

Comments and Suggestions for Authors

  • Provide more details on the experimental setup and methodology used to evaluate the salt adaptation strategies of the two grape rootstocks. Include information on the growth conditions, salinity treatments, and duration of the experiments. This would enhance the reproducibility of the study and allow readers to better understand the experimental design.

  • The section was elaborated accordingly.

  • Introduction needs to be benefited from more literature, and papers have to be cited in it, i.e., I) doi: 10.1128/spectrum.02311-22 II) doi: 10.1039/d2gc02467e.

  • We fully appreciate your recommendation. However, they are a little bit far from our scope as they deal with bacteria, Two-Component Systems, and other stress. Of course, they are helpful, but kindly, we try to be more specific and focused. Again, we highly evaluate your suggestions.

  • Include additional physiological parameters to assess the salt adaptation of the rootstocks. For example, measuring the activity of antioxidant enzymes such as superoxide dismutase (SOD) and catalase (CAT) would provide insights into the ROS detoxification machinery. Additionally, measuring ion content (Na+, Cl-, etc.) and osmolyte accumulation (e.g., proline) would further elucidate the mechanisms underlying salt tolerance.
  • We already measured the activity of antioxidant enzymes including SOD, CAT, and GPX, as well as the content of proline. For ions content such as Na and Cl, unfortunately, we did not have enough sample materials to measure them, as well as the funds were already run out.

  • Discuss the implications of the observed differences in photosynthetic pigments and sugar accumulation between the two rootstocks. How do these differences contribute to their respective salt adaptation strategies? Elaborate on the potential influence of these physiological traits on plant growth, development, and overall salt tolerance. Discussion needed to be more comparatively from previous literature i.e., I) https://doi.org/10.3390/polym14142899.

We fully appreciate your suggestions, and we worked throughout all parts of the manuscript, particularly, the discussion part. 

  • Consider including gene expression analysis or proteomic profiling to investigate the molecular mechanisms underlying the observed physiological responses. This would provide a more comprehensive understanding of the molecular pathways involved in salt adaptation and enable the identification of specific genes or proteins associated with the differential salt tolerance of the rootstocks.

  • We fully agree with your suggestions. So, we are currently analyzing the RNAseq data for salt-stressed RUG, as well as linking these physiological data with the transcriptomic data to further understand the underlying mechanisms. To make the story easier, we decided to divide the results into two manuscripts; physiological and transcriptomic. However, we are confident that the physiological story is connected and informative, although the transcriptomic data that will be published later will provide further and comprehensive information.

  • Discuss the practical implications of the findings in the context of grape cultivation in salt-affected regions. Highlight the potential benefits of using salt-excluder rootstocks like RUG in improving grapevine productivity and sustainability under salinity stress. Additionally, discuss the limitations and challenges associated with the adoption of these rootstocks in commercial vineyards.

We fully appreciate your suggestions, and we worked throughout all parts of the manuscript, to provide all necessary information. 

Round 2

Reviewer 1 Report

plants-2541138

Title : Physiological Comparison of Two Salt-Excluder Hybrid Grapevine Rootstocks Under Salinity Reveals Differential Adaptation Quality

I am sorry, despite these efforts, I found the responses of authors to my previous critical comments not convincing. In its current state, the article is merely descriptive in nature, lacking in clear hypotheses and mechanistic insight, and does not add anything new to scientific knowledge, and thus does not fall within the level and standards of the journal.

Moderate editing of English language required